

# Marine amphipods (*Parhyale hawaiensis*) as an alternative feed for the lined seahorse (*Hippocampus erectus*, Perri 1810): nutritional value and feeding trial

Jorge Arturo Vargas-Abúndez[1], Gemma Leticia Martínez-Moreno[2], Nuno Simões[3,4], Elsa Noreña-Barroso[3,5] and Maite Mascaró[2,3]

[1] Posgrado en Ciencias del Mar y Limnología, Universidad Nacional Autónoma de México, Mexico city, Mexico
[2] Unidad Multidisciplinaria de Docencia e Investigación (UMDI-Sisal), Facultad de Ciencias, Universidad Nacional Autónoma de México, Sisal, Yucatán, Mexico
[3] Laboratorio de Resiliencia Costera (LANRESC, CONACYT), Sisal, Yucatán, Mexico
[4] International Chair for Coastal and Marine Studies in Mexico, Harte Research Institute for Gulf of Mexico Studies, Texas A&M University-Corpus Christi, *Corpus* Cristi, Texas, United States of America
[5] Unidad de Química en Sisal, Facultad de Química, Universidad Nacional Autónoma de México, Sisal, Yucatán, Mexico

Corresponding author
Maite Mascaró,
mmm@ciencias.unam.mx

## ABSTRACT

Finding new alternatives to traditional live preys such as *Artemia* and rotifers, which do not always promote optimal fish growth and survival, is required for the successful aquaculture of highly specialized predatory species, including seahorses. The present study assessed the nutritional value of an interesting marine amphipod (*Parhyale hawaiensis*), and evaluates through a feeding trial its potential use as a natural prey for 10-months lined seahorse, *Hippocampus erectus*. *P. hawaiensis* showed high levels of valuable lipids (20.4–26.7% on dry matter basis) and polyunsaturated fatty acids (PUFAs) ( 26.4–41% of total FAs), including the long-chain PUFAs (LC-PUFAs) arachidonic acid (ARA) (2.9–7.7%), eicosapentaenoic acid (EPA) (4.3–6.5%) and docosahexaenoic acid (DHA) (2.1–6.2%). A comparison between wild-captured and cultured amphipods revealed a significant improvement of the amphipod FA profile in terms of DHA%, total omega-3 (n3) FAs and n3/n6 ratio when employing both a conventional amphipod culture based on a commercial shrimp diet, and, to a lesser extent, a large (3,500 L) biofloc system. Seahorses fed with frozen/wild amphipods, either singly or in combination with *Artemia* enriched with Super Selco® (INVE Aquaculture, Belgium) for 57 days, substantially improved seahorse growth and FA profiles in terms of ARA, EPA and DHA%, including indices associated to marine sources, such as Σn3 and n3/n6, compared to a diet based solely on enriched *Artemia*. These results support the use of marine amphipods as an alternative food organism for juvenile *H. erectus* and suggest a potential use for general marine aquaculture.

## INTRODUCTION

Live food organisms are indispensable for the early culture of many marine species of commercial interest, including marine ornamentals (*Olivotto et al., 2011*, *2017*; *Southgate, 2019*). Due to the predatory nature of some species, live food stimulates a better feeding response compared to inert feeds, in addition to being more easily digested and assimilated (*Conceição et al., 2010*). The ideal live foods are fundamentally the preys a particular species encounter in nature and these often include small crustaceans such as copepods, amphipods and decapods (*Olivotto, Planas & Turchi, 2017*). The production of natural food organisms is, unfortunately, laborious and expensive, at best, and unsuccessful for most species (*Southgate, 2019*). The capture of live food organisms from the wild provides a viable alternative to captive propagation. Notwithstanding its cost-effectiveness, this practice is subject to seasonal availability and susceptible to the undesired introduction of pathogens and pests (*Cohen & Valenti, 2019*).

Because of their relatively easy production and cost, *Artemia* sp. and rotifers are the most commonly used live food organisms (*Southgate, 2019*). These preys are not the natural prey of marine fish, but they are widely used and with relative success (*Bengtson, 2003*). One of their main limitations is that they do not always satisfy the nutritional requirements of all organisms (*Sorgeloos, Dhert & Candreva, 2001*). "Enriching" these preys with formulations such as oil emulsions rich in essential FAs (EFAs) or microalgae, may overcome nutritional deficiencies; but even implementing this costly practice, they are often inadequate for the culture of many species, including seahorses (*Segade et al., 2016*; *Randazzo et al., 2018*; *Planas et al., 2020*).

In the last years, marine amphipods have received increasing attention as an alternative natural food. These benthic crustaceans can form large colonies (>100,000 individuals m$^{-2}$) in natural or artificial aquatic habitats, such as coral reefs, seagrasses, seaweeds and biofoulings (*Lourido, Moreira & Troncoso, 2008*; *Vázquez-Luis, Sanchez-Jerez & Bayle-Sempere, 2013*; *Navarro-Mayoral et al., 2020*), where they constitute a major natural prey of small fish (<30 cm standard length) and invertebrates (*Woods, 2009*). They are promising candidates both for intensive and extensive culture, as they can feed on a variety of foodstuffs, including decaying organic material and detritus (*Guerra-García et al., 2016*), and tolerate wide ranges of environmental parameters, such as temperature (>20 °C range) and salinity (>20 psu range) (*Takeuchi, Matsumasa & Kikuchi, 2003*; *Campbell et al., 2020*). Although diet and environmental parameters can influence their nutritional value, they are generally rich in proteins (up to 60% of dw) and lipids (up to 20% on dry weight (dw) basis), including the PUFAs (over 50% of total FAs in some species) EPA (over 20% of total FAs) and DHA (up to 20% of total FAs) (*Wang & Jeffs, 2014*; *Fernandez-Gonzalez et al., 2018*; *Jiménez-Prada et al., 2018*). The few studies to assess and develop potential culturing or harvesting techniques for aquaculture conducted so far are promising (*Guerra-García et al., 2016*; *Fernandez-Gonzalez et al., 2018*; *Xue et al., 2018*; *Vargas-Abúndez et al., 2021*). These include a recently published culture trial in biofloc systems (*Promthale et al., 2021*).

Feeding trials to assess the potential use of amphipods as alternative feed conducted so far are encouraging. As a partial fishmeal replacement, the Arctic amphipod (*Themsto*

*libellula*) was successfully incorporated into the diets of Atlantic salmon (*Salmo salar*) and Atlantic halibut (*Hippoglossus hippoglossus*), substituting 40% of fishmeal in diet (*Suontama et al., 2007*). *Gammarus* species have also been of interest as amphipod meal (*Harlıoğlu & Farhadi, 2018*). Both as live and frozen feed, several amphipod species have been successfully used to replace traditional live foods in the culture of seahorses (*Murugan et al., 2009*; *Vargas-Abúndez, Simões & Mascaró, 2018*), octopus (*Baeza-Rojano et al., 2013b*) and cuttlefish (*Baeza-Rojano et al., 2010*).

According to traditional classification (*Martin & David, 2001*), *Parhyale hawaiensis* (Dana, 1853) is a gammarid amphipod with a worldwide, circumtropical distribution (*WoRMS Editorial Board, 2020*). It inhabits marine coastal habitats such as rocky beaches, estuaries and mangroves, where it readily forms dense aggregations of up to 7,000 individuals m$^{-2}$ (*Poovachiranon, Boto & Duke, 1986*; *Paz-Ríos, Simões & Ardisson, 2013*). Since the early 2000's, it has been attracting interest to the scientific community as a compelling crustacean model for biological research (*Sun & Patel, 2019*), due to their small size (6.12–11.83 mm, total length), fast growth (0.15 mm day$^{-1}$), short life cycle (from newborn to adult in 50.9 ± 5.8 days), high fecundity (up to 35 embryos per female), translucent embryos and year-round reproduction (*Vargas-Abúndez et al., 2021*). It is amenable to experimental investigation and plenty of information and experimental tools, such as the complete genome and gene editing tools, are already available for this species (*Kao et al., 2016*; *Sun & Patel, 2019*). Propagation of *P. hawaiensis* under laboratory conditions is straightforward and well documented, with prospects for mass-scale aquaculture (*Vargas-Abúndez et al., 2021*). Taking advantage of its environmental tolerance, opportunistic behavior and detritivorous habits, *P. hawaiensis* could be an interesting candidate for mass production in biofloc systems. Biofloc technology represents an innovative approach for environmental-friendly, cost-effective intensive farming (*Avnimelech, 2015*; *Ahmad et al., 2017*).

The natural diet of most seahorse species is dominated by small crustaceans, primarily copepods, mysid shrimps, decapods and amphipods (*Manning, Foster & Vincent, 2019*). However, amphipods stand out as the main food item in terms of frequency of occurrence, number or biomass ingestion for species such as *H. breviceps*, *H. coronatus*, *H. erectus*, *H. guttulatus*, *H. hippocampus*, *H. patagonicus*, *H. subelongatus* and *H. whitei* (*Teixeira, Musick & Musik, 2001*; *Kendrick & Hyndes, 2005*; *Kitsos et al., 2008*; *Storero & González, 2008*). Adult *H. erectus*, the lined seahorse, feed almost exclusively on amphipods (mainly *Gammarus muconathus*), whereas juveniles on both amphipods (mainly *Ampithoe longimana*) and copepods (*Teixeira, Musick & Musik, 2001*). Feeding trials conducted by the authors demonstrated that this heavily traded species showed an increased feeding response when fed with frozen amphipods (*Elasmopus pectenicrus*) compared to live *Artemia* (*Vargas-Abúndez, Simões & Mascaró, 2018*), confirming the potential of amphipods to overcome one of the main bottlenecks of seahorse aquaculture, adequate feeding (*Koldewey & Martin-Smith, 2010*). Whether seahorse aquaculture can meet the goal of providing a sustainable alternative to supply the traditional medicine, aquarium and curio industries, highly depends on developing adequate live prey foods (*Koldewey & Martin-Smith, 2010*).

Considering the aforementioned, the aim of the present study was (a) to assess the nutritional value of *P. hawaiensis* in relation to lipids and fatty acids (FA), and (b) to test their effects as a full or supplemental diet on growth, survival and FA profiles of juvenile *H. erectus*. Wild-captured amphipods were used in seahorse feeding trial. Additionally, and for the first time, amphipods were produced in a large-scale biofloc system and their nutritional value explored. The present study represents the first one evaluating the possible use of amphipods for seahorse feeding, testing the actual value of a new and promising amphipod species.

## MATERIALS & METHODS

### Ethics

The present study was carried out under a permit by SEMARNAT No. SGPA/DGVS/ 12741/13 and strictly followed institutional protocols for the maintenance, manipulation, and sacrifice of the experimental animals according to certified criteria established by the Guide for the Care and Use of Experimental Animals in Research and Teaching of the Faculty of Superior Studies-Cuautitlán (http://www.cuautitlan.unam.mx/) at Universidad Nacional Autónoma de México. During the experiment, seahorse mortality was kept at zero and no apparent signs of stress were detected (*i.e.*, changes in color, disease, lack of feeding or mobility).

### Foods of different source

Amphipods (*P. hawaiensis*) were collected as previously described in *Vargas-Abúndez et al. (2021)*. Specifically, they were collected both from outdoor flow-through systems in which amphipods grow freely at aquaculture facilities of the National Autonomous University of Mexico (UNAM) located in Sisal, Yucatán, México, and from green intertidal algae attached to rocks in Sisal beach. They are abundant in Sisal beach, from where they likely infiltrate the aquaculture systems. The animals were rinsed with fresh water and then some were immediately frozen at $-80\ ^\circ$C for further lipid content and FA analysis, and at $-18\ ^\circ$C (commercial freezer) for use in the feeding trial. Hereafter, these amphipods are referred to as captured amphipods. The rest of the amphipods were acclimatized to laboratory conditions and held in an in-door 250 L tank with gentle aeration. Water in the tank was partially changed twice a week and maintained at $29.1 \pm 2.7\ ^\circ$C, salinity $35.9 \pm 5.5$ ppt, pH $8.1 \pm 0.4$, $NO_2^-$ $1.3 \pm 3.4$ mg $L^{-1}$, $NO_3^-$ $24.8 \pm 26.1$ mg $L^{-1}$, $NH_3/NH_4^+$ $0.6 \pm 1.4$ mg $L^{-1}$. Plastic mesh was introduced in the tank as artificial substratum for the animals (*Baeza-Rojano et al., 2013a*; *Vargas-Abúndez et al., 2021*). Amphipods were fed daily with a commercial shrimp feed (Camaronina 35® Purina, Sonora, Mexico) (crude protein 350 g $kg^{-1}$, lipids 80 g $kg^{-1}$, ash $<$ 100 g $kg^{-1}$, fiber $<$ 50 g $kg^{-1}$, energy 21.6 kJ $g^{-1}$, FA profile not available). Amphipods maintained this way were also used in the feeding trial and were labelled pellet-fed amphipods.

Simultaneously, amphipods were cultured in a biofloc system, following a modified protocol for shrimp biofloc systems described in *Magaña-Gallegos et al. (2018)*, with some modifications. The culture took place in a large, out-door tank (3,500 L) exposed to coastal climate conditions from May to December. Water in the tank was maintained at

$27.2 \pm 2.1\ ^{\circ}$C, salinity $28.6 \pm 7.3$ ppt, pH $8.4 \pm 0.1$, $NO_2^-$ $0.7 \pm 0.9$ mg $L^{-1}$, $NO_3^-$ $18.8 \pm 19.4$ mg $L^{-1}$, $NH_3/NH_4^+$ $0.6 \pm 1.1$ mg $L^{-1}$. To stimulate the growth of nitrite-oxidizing bacteria, sodium nitrite ($NaNO_2$) was added at the beginning of the culture (*Lara et al., 2016*). To promote the generation of bioflocs, sugarcane molasses and wheat bran were added as carbon sources (*Avnimelech, 2015*). These were added at the beginning of the culture trial and every 2 weeks thereafter, until the biofloc volume reached five ml $L^{-1}$ (biofloc volume was measured with Imhoff cones). When the biofloc volume decreased to less than five ml $L^{-1}$ or total ammonia nitrogen (TAN) reached one mg $L^{-1}$, carbon addition was resumed (*Emerenciano et al., 2013*). A high C/N ratio of 20/1 was maintained, and water exchange was limited to compensate for evaporation. Sludge was removed occasionally from the tank by a central drain. The water was continuously aerated and pieces of plastic mesh were introduced as substrate for the amphipods. *P. hawaiensis* amphipods were introduced into the tank 3 weeks after the beginning of the culture. A commercial shrimp feed (Camaronina 35® Purina, Sonora, Mexico) was administered three times a week in excess, as an additional source of nitrogen and supplemental feed for the amphipods. This experimental group was labeled biofloc amphipods.

*Artemia (ProAqua®, Sinaloa, Mexico)* was raised with wheat bran during the first 16 days and then, the last 6–8 days, with *Spirulina* sp. Prior to its use, it was enriched with Super Selco® (INVE Aquaculture, Dendermonde, Belgium) in one L tanks for six hours at a concentration of six mL $L^{-1}$. Enrichment period was chosen to avoid both FA autoxidation and FA retroconversion by *Artemia* (*McEvoy et al., 1995*; *Nieves-Soto et al., 2021*). Supporting this choice, preliminary observations indicated a significant increase in EPA and DHA percentage within six hours, as it was further confirmed by results herein (see "Foods of different source").

## Seahorses

Wild pregnant *H. erectus* Perry, 1810 were captured at Laguna de Chelem, Yucatán, Mexico under a scientific license (SGPA/DGVS/12741/13) from the Mexican Ministry of the Environment and Natural Resources (SEMARNAT). Fish maintenance followed previously published methods by *Vargas-Abúndez, Simões & Mascaró (2018)*. After birth, juveniles were maintained in re-circulating holding tanks (30 H × 28 L × 18 W cm, 14 L). Seawater was treated with mechanical (25, 10 and 5 µm), biological and UV filtration. Water in the aquaria was maintained at 26. $\pm$ 0.5 $^{\circ}$C (mean $\pm$ standard deviation), salinity $36.4 \pm 2.5$ psu, pH 8.0–8.3, $NO_2^- < 0.3$ mg $L^{-1}$, $NO_3^- < 5$ mg $L^{-1}$, $NH_3/NH_4^+ < 0.1$ mg $L^{-1}$ with a gentle aeration. A 12:12 photoperiod was kept throughout experiments. Polypropylene structures were placed in the aquaria to be used as holdfasts by the fish. Juvenile fish were fed three times a day (09:00 h, 14:00 h, 18:00 h) with live and frozen *Artemia* enriched with Super Selco® (INVE Aquaculture, Dendermonde, Belgium). At 50 mm standard length, fish were weaned from *Artemia* to frozen amphipods and then fed with a mix of the two foods, according to previous findings and culture recommendations (*Lin et al., 2009*; *Vargas-Abúndez, Simões & Mascaró, 2018*; *Del Vecchio et al., 2019*). Feces and uneaten food were siphoned out after feeding.

## Seahorse feeding trial

Forty-eight *H. erectus* juveniles (21 males and 27 females) of ca. 10 months old (1.3 ± 0.4 g wet weight, ranging from 0.72–2.48 g) were randomly selected and individually tagged with a collar tag (*Morgan & Bull, 2005*). Fish were divided into 12 tanks of 15 L (30 cm × 20 cm × 30 cm) with four fish in each tank, which were in turn evenly and randomly assigned to one of the three following dietary treatments (four tanks per dietary group) (all diets frozen): (i) amphipod diet: 100% captured amphipods; (ii) *Artemia* diet: 100% enriched *Artemia*; (iii) mixed diet: a 1:1 mix of the captured amphipods and the *Artemia* diets.

Fish were fed in excess (25% of wet body weight per day) three times a day for 57 days. Water characteristics were maintained as previously described for seahorses. Fish growth was assessed through individual wet weight, which was repeatedly measured in each individual at the beginning of the experiment (day 0) and at days 15, 30, 45 and 57 with an OHAUS Adventurer analytical balance. Fish growth among experimental groups was not influenced by animal gender, as shown by data exploration (for raw data see Supplementary Materials). For comparative purposes, the specific growth rate (SGR) for individuals of each experimental group was calculated as follows:

$$SGR\% = ((\ln Wf - \ln Wi)/t) \times 100$$

where Wf is the final wet weight, Wi, the initial wet weight, and t, the number of days. Survival was recorded daily. At the end of the experiment (day 57) fish were euthanized by quick submersion in a mix of ice and water (hypothermia) and stored at −80 °C for further FA acid analysis.

## Lipid content and fatty acid analysis

To assess the nutritional value of amphipods, lipid content and FA analyses were conducted on samples of captured, biofloc (which treatment included shrimp food) and exclusively pellet-fed amphipods. As a control group to the FA analyses, enriched *Artemia* was included. Once harvested from the different sources, amphipod samples were rinsed with freshwater and sieved through a 710 μm mesh; retained juveniles and adults were used for the analyses.

Samples were minced, freeze-dried and homogenized in liquid nitrogen with a commercial blender. Lipid extraction was carried out based on Folch extraction procedure with dichloromethane/methanol (2:1 v/v) (*Folch, Lees & Sloane-Stanley, 1987*). Extracts were saponified with 20% KOH:Methanol (w/v) and FAs were obtained from the saponifiable fraction (pH = 1–2) using hexane as solvent. FAs were esterified with 10% BF3 in methanol (Fluka 15,716) for 60 min at 80 °C and FA methyl esters (FAME) were obtained. FAME were separated and quantified by gas chromatography using a Perkin Elmer Clarus 500 gas chromatograph (GC) equipped with a flame ionization detector (FID), and a Phenomenex Zebron ZB-WAX capillary column (20 m length, 0.18 mm i.d., 0.18 μm film thickness). Hydrogen was used as the carrier gas at a flow rate of 40 mL min$^{-1}$. The column temperature was programmed to increase from 40 to 200 °C at a rate of 20 °C min$^{-1}$ and from 200 to 250 °C at a rate of 2.5 °C min$^{-1}$, whereas injector and

detector temperatures were set at 280 and 250 °C, respectively. Individual components were identified according to their retention times using analytical standards (Supelco® 37 Component FAME Mix, catalog no. 47885-U) as reference. Individual FA concentrations were expressed as percentages of the total FA composition.

Fatty acid determinations were performed on seahorses once at the end of the trial. Six whole fish samples per experimental group were randomly selected from previously sacrificed fish, and analyzed in duplicate. Samples were minced, freeze-dried and homogenized in liquid nitrogen with a commercial blender prior to analysis. In order to compare the FA profile of samples from different foods and seahorses, the following indices were calculated: total saturated FAs (ΣSFA), total monounsaturated FAs (ΣMUFA), total polyunsaturated FAs (ΣPUFA), n3 highly-unsaturated FAs (n3 HUFA; = C20:3n3 + C20:5n3 + C22:6n3), total n3 FAs (Σn3), total n6 FAs (Σn6), n3/n6, DHA/EPA and EPA/ARA.

### Statistical analysis

Statistical differences in total lipid content among amphipods of different source (captured amphipods ($n = 2$), biofloc amphipods ($n = 3$) and pellet-fed amphipods ($n = 3$)) were analyzed by a one-way ANOVA. The statistical software package Prism5 (GraphPad Software) was used for this analysis.

Variations in the FA composition between sources of food and seahorses fed with three different diets were assessed by means of Principal Coordinate Analyses (PCoA). Whitaker's association index ($D_9$; *Legendre & Legendre, 1998*; *Borcard, Gillet & Legendre, 2011*) was applied to the data, expressed as proportions of the total FA content in each sample in order to obtain a resemblance matrix with dissimilarity measures between every pair of samples. Non-metric Dimensional Scaling (nMDS) was used to compare the index values calculated to characterize the FA profiles of both foods and seahorses. In this case, the Gower coefficient ($S_{15}$; *Legendre & Legendre, 1998*) was used to calculate multivariate distances between samples. Both the 2D and 3D configurations were obtained together with Kruskall's stress coefficient (*Clarke, Gorley & Somerfield, 2014*) and the best was selected on the basis of stress criteria described in *Legendre & Legendre (1998)*.

Multiple ANOVAs with permutations (*Anderson, 2001*) were used to distinguish differences in FA composition and indices related to food source and seahorse diet from random noise. In the first case, the underlying model was a one-way ANOVA with food source as a fixed factor with four levels: captured amphipods ($n = 4$), biofloc amphipods ($n = 3$), pellet-fed amphipods ($n = 3$) and enriched *Artemia* ($n = 2$). The underlying model in the second case had seahorse diet as a fixed factor with three levels (amphipod, *Artemia* and a mixed diet), individual seahorses as a random factor nested within each level diet (b = 6), and $n = 2$ replicate subsamples of every individual. Permutations of residuals under the reduced model (9,999) were used to generate empirical distributions of *pseudo-F* values under the null hypotheses (*Anderson, 2017*). *Post hoc* comparisons were applied following a similar procedure after the main test indicated significant differences ($p < 0.05$) between at least two centroids. Multivariate procedures were carried out using PRIMER 7 and PERMANOVA + for PRIMER.

Changes in seahorse wet weight through time was evaluated through regression analysis adjusting a mixed linear model (GLMM) with diet as a fixed factor (three levels: amphipod diet, *Artemia* diet and mixed diet) and time (days) as a continuous variable. Preliminary data exploration showed that data did not comply to homoscedasticity (*i.e.*, dispersion in seahorse weight increased as mean weight increased) or independence (*i.e.*, seahorse weight was repeatedly measured on individuals through time). To ensure the reliability on the estimated coefficients and standard errors and *p*-values obtained (*Zuur, Leno & Smith, 2007*), the model was adjusted with a generalized least-square procedure through restricted maximum likelihood and incorporated correlation and variance structures. The intercepts and slopes of linear equations corresponding to the three diets were compared with t-tests using the residual standard error estimated by the model. Different slopes would indicate different seahorse growth rates (mg day$^{-1}$), irrespective of seahorse initial weights. The goodness of fit of the model was validated by visual inspection of residuals (*Montgomery & Peck, 1992*; *Zuur, Leno & Smith, 2007*). The R libraries nlme (*Pinheiro, Bates & DebRoy, 2020*) and ggplot2 (*Wickham, 2016*) were used to adjust the GLMM and generate the graphic visualization.

## RESULTS

### Foods of different source

Biofloc and pellet-fed amphipods showed higher total lipid contents (%dw) (26.7 ± 1.3% and 25.5 ± 3.5% lipids, respectively) compared to captured amphipods (20.4 ± 0.8%), yet these were not statistically significant ($F = 4.56$; $p = 0.07$). Table 1 shows the FA composition of all food sources. Multivariate analysis on the FA composition of food sources showed an effective reduction of dimensionality with the first and second principal coordinates containing 80% of the total variation in the data (Fig. 1; Table S1). Samples from captured amphipods were located to the right-hand side of the ordination map and were associated with high contents of oleic (C18:1n9c/t), arachidonic (C20:4n6), dihomo-gamma-linoleic (C20:3n6), pentadecylic (C15:0), lauric (C12:0) and margaric (C17:0) acids. By contrast, samples from *Artemia*, pellet-fed and biofloc amphipods were high in docosahexaenoic (C22:6n3), stearic (C18:0) and docosadienoic (C22:2) acids and were located to the left-hand side of the map (Fig. 1). *Artemia* and biofloc amphipods had the highest contents of linoleic (C18:2n6c), palmitoleic (C16:1) and alpha-linolenic (C18:3n3) acids, followed by captured and pellet-fed amphipods (see Table S1 for details on the contribution of each descriptor to the linear combinations of the first three principal coordinates).

Results of the MANOVA revealed significant differences in FA composition of foods related to its source (*pseudo-F = 7.51*; $p < 0.001$; 9,626 unique permutations; Table 2), and clearly separated wild-captured amphipods from biofloc and exclusively pellet-fed amphipods (*pseudo-F = 2.54 and 3.17*; $p < 0.05$; 45 unique permutations, respectively). However, samples from *Artemia* could not be statistically distinguished from any of the other groups (*pseudo-F from 2.53 to 3.5*; $p$ from 0.07 to 0.1; 10 to 15 unique permutations), probably due to its low number of replicates ($n = 2$; Table 2).

**Table 1 Fatty acid composition (as percentage of total FAs) of *P. hawaiensis* of different source and enriched Artemia.**

| FAs | Captured amphipods | Biofloc amphipods | Pellet-fed amphipods | *Artemia* |
|---|---|---|---|---|
| C12:0 | 0.4 ± 0.21 | 0.74 ± 0.08 | 0.81 ± 0.12 | 0.05 ± 0.01 |
| C13:0 | 0.06 ± 0 | 0.06 ± 0.01 | 0.06 ± 0.01 | 0.04 ± 0 |
| C14:0 | 5.74 ± 1.62 | 7.36 ± 0.68 | 8.59 ± 0.91 | 1.71 ± 0.18 |
| C14:1 | 0.04 ± 0 | 0.05 ± 0 | 0.04 ± 0 | 0.01 ± 0 |
| C15:0 | 0.97 ± 0.06 | 0.78 ± 0.06 | 0.59 ± 0.19 | 0.64 ± 0.07 |
| C16:0 | 22.49 ± 7.96 | 15.75 ± 5.59 | 14.97 ± 0.69 | 17.47 ± 1.53 |
| C16:1 | 5.71 ± 6.32 | 6.44 ± 1.67 | 4.86 ± 3.14 | 9.22 ± 1.08 |
| C17:0 | 2.35 ± 0.4 | 1.3 ± 0.08 | 0.34 ± 0.13 | 2.43 ± 0.19 |
| C17:1 | 0.54 ± 0.05 | 0.23 ± 0.14 | 0.45 ± 0.27 | 0.14 ± 0.02 |
| C18:0 | 8.23 ± 2.64 | 12.66 ± 4.12 | 13.89 ± 2.71 | 9.41 ± 0.7 |
| C18:1n9c/t | 22.9 ± 1.44 | 18.23 ± 5.21 | 8.6 ± 3.34 | 15.71 ± 2.81 |
| C18:2n6c | 5.58 ± 0.94 | 14.8 ± 2.11 | 7.04 ± 2.99 | 20.78 ± 1.06 |
| C18:3n6 | 1.19 ± 0.77 | 0.17 ± 0.04 | 5.51 ± 1.76 | 0 ± 0 |
| C18:3n3 | 0.65 ± 0.7 | 2.12 ± 0.12 | 1.92 ± 1.04 | 2.23 ± 0.09 |
| C20:0 | 0.43 ± 0.12 | 0.44 ± 0.06 | 0.68 ± 0.05 | 0.27 ± 0.03 |
| C20:1n9 | 2.34 ± 0.18 | 3.22 ± 0.18 | 4.12 ± 1.06 | 1.58 ± 0.07 |
| C20:2 | 0.8 ± 0.14 | 3.06 ± 0.14 | 4.19 ± 0.17 | 0.45 ± 0.01 |
| C20:3n6 | 0.74 ± 0.33 | 0.19 ± 0.03 | 0.25 ± 0.01 | 0.05 ± 0.03 |
| C21:0 | 0.17 ± 0.01 | 0.12 ± 0 | 0.15 ± 0.01 | 0.02 ± 0 |
| C20:3n3 | 0.35 ± 0.04 | 0.65 ± 0.03 | 0.81 ± 0.04 | 0.03 ± 0.01 |
| C20:4n6 | 7.68 ± 2.62 | 2.92 ± 0.13 | 2.98 ± 0.12 | 4.97 ± 0.35 |
| C20:5n3 | 6.63 ± 0.53 | 4.26 ± 0.2 | 5.65 ± 0.23 | 6.94 ± 0.95 |
| C22:0 | 0.45 ± 0.06 | 0.22 ± 0.04 | 0.32 ± 0.03 | 0.49 ± 0.12 |
| C22:1n9 | 0.34 ± 0.03 | 0.23 ± 0.06 | 0.35 ± 0.05 | 0.19 ± 0.03 |
| C22:2 | 0.09 ± 0.08 | 0.12 ± 0.02 | 0.18 ± 0.01 | 0.08 ± 0.09 |
| C23:0 | 0.21 ± 0.07 | 0.11 ± 0.03 | 0.13 ± 0.02 | 0.04 ± 0.02 |
| C24:0 | 0.25 ± 0.16 | 0.05 ± 0.01 | 0.09 ± 0 | 0.08 ± 0.02 |
| C22:6n3 | 2.66 ± 0.71 | 3.72 ± 0.19 | 6.17 ± 0.25 | 4.72 ± 1.33 |
| ΣSFA | 41.74 ± 6.19 | 39.59 ± 4.79 | 40.61 ± 1.53 | 32.65 ± 2.84 |
| ΣMUFA | 31.87 ± 5.97 | 28.4 ± 3.42 | 18.41 ± 6.18 | 26.84 ± 1.6 |
| ΣPUFA | 26.38 ± 2.35 | 32.01 ± 1.43 | 34.7 ± 3.71 | 40.22 ± 1.34 |
| n3 HUFA | 9.64 ± 0.94 | 8.63 ± 0.42 | 12.63 ± 0.49 | 11.66 ± 2.28 |
| Σn3 | 10.29 ± 1.46 | 10.75 ± 0.53 | 14.55 ± 1.49 | 13.89 ± 2.19 |
| Σn6 | 15.2 ± 2.96 | 18.07 ± 1.98 | 15.78 ± 2.17 | 25.8 ± 0.74 |
| n3/n6 | 0.71 ± 0.21 | 0.6 ± 0.1 | 0.93 ± 0.06 | 0.54 ± 0.1 |
| DHA/EPA | 0.4 ± 0.12 | 0.87 ± 0 | 1.09 ± 0.02 | 0.67 ± 0.1 |
| EPA/ARA | 0.95 ± 0.34 | 1.46 ± 0.03 | 1.9 ± 0.01 | 1.39 ± 0.09 |

**Note:**
Data show ± one standard deviation of the mean.

The 3D nMDS configuration map of several indices describing the FA profiles of food sources had a stress coefficient of 0.02. A 2D projection of this configuration showed that captured amphipods had high values of ΣSFA and ΣMUFA, followed by biofloc and

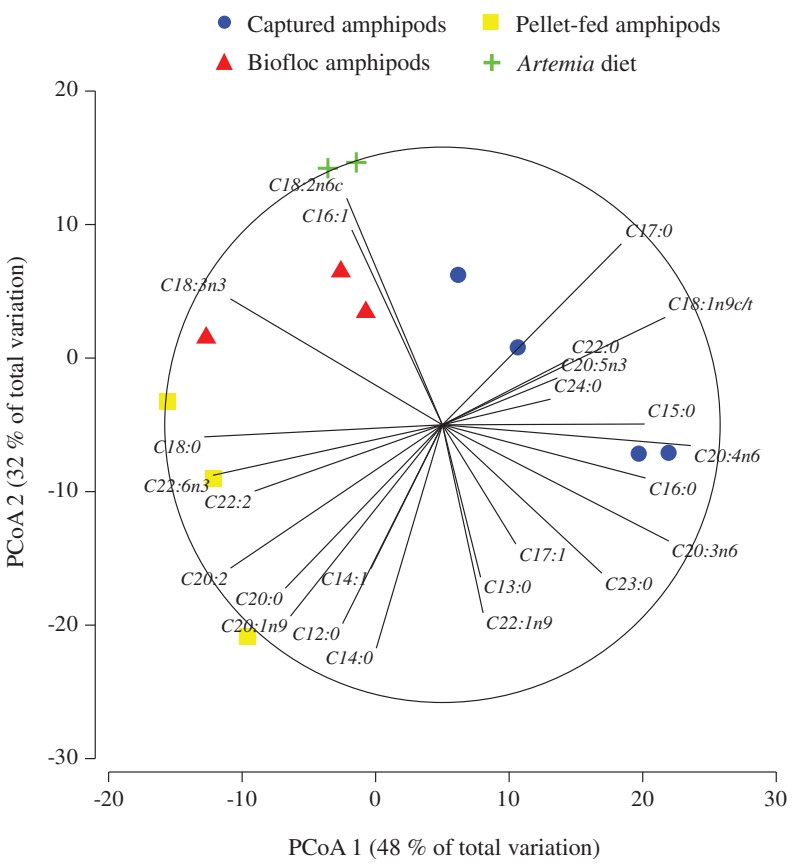

**Figure 1 Principal coordinate analysis of fatty acid composition of four diets elaborated with amphipod _P. hawaiensis_ of different source and enriched _Artemia_.**

pellet-fed amphipods (Fig. 2). _Artemia_ samples had the highest values of ΣPUFA and Σn6, whereas pellet-fed amphipods had the highest values of EPA/ARA and Σn3 (Fig. 2; see Table S2 for details on the multiple correlation coefficients of descriptors with the first three ordinal axes). Here again, the MANOVA procedures statistically distinguished captured amphipods from biofloc and pellet-fed amphipods (_pseudo-F_ = 1.78 and 3.56; $p < 0.05$; 35 unique permutations, respectively), but was unable to find significant differences between the latter (_pseudo-F_ = 4.19; $p = 0.09$; 10 unique permutations, Table 3). _Artemia_ samples were again statistically similar to all the other groups (_pseudo-F_ from 2.49 to 3.64; p from 0.07 to 0.1; 10 to 15 unique permutations; Table 3).

## Seahorses

All the animals used for the present study survived until the end of the experiment. The GLMM showed a significant interaction term indicating that linear equations describing fish growth differed depending on the diet ($F = 46.2$; $p < 0.0001$). Further comparisons of the regression coefficients showed statistically similar intercepts but different slopes in the corresponding linear equations (Table 4). Results demonstrated that captured amphipods in the diet, both solely (amphipod diet: 13 ± 2 mg day$^{-1}$; 0.8 ± 0.3%

**Table 2 Results of a permutational MANOVA applied on the fatty acid composition of four diets elaborated with amphipod *P. hawaiensis* of different source and enriched *Artemia*.**

| Source of variation | df | SS | MS | pseudo-F | p | Unique permutations |
|---|---|---|---|---|---|---|
| Food source | 3 | 2617.2 | 872.4 | 7.5 | <0.001 | 9,626 |
| Residual | 8 | 929.3 | 116.2 | | | |
| Total | 11 | 3546.5 | | | | |
| **Post-hoc comparisons** | | | | **pseudo-t** | **p** | **Unique permutations** |
| Wild amphipods *vs* Biofloc amphipods | | | | 2.5 | <0.05 | 35 |
| Wild amphipods *vs* Pellet amphipods | | | | 3.2 | <0.05 | 35 |
| Biofloc amphipods *vs* Pellet amphipods | | | | 3.5 | 0.10 | 10 |
| Artemia *vs* Biofloc amphipods | | | | 2.3 | 0.11 | 10 |
| Artemia *vs* Pellet amphipods | | | | 3.5 | 0.10 | 10 |
| Artemia *vs* Wild amphipods | | | | 2.5 | 0.07 | 15 |

**Note:**

df, degrees of freedom; SS multivariate sums of squares; multivariate mean squares; *pseudo-F*, *pseudo-t* and *p*: F and t values obtained through permutations of the reduced model and the *p* values associated; number of unique permutations used to obtain each *pseudo-F* and *pseudo-t* value.

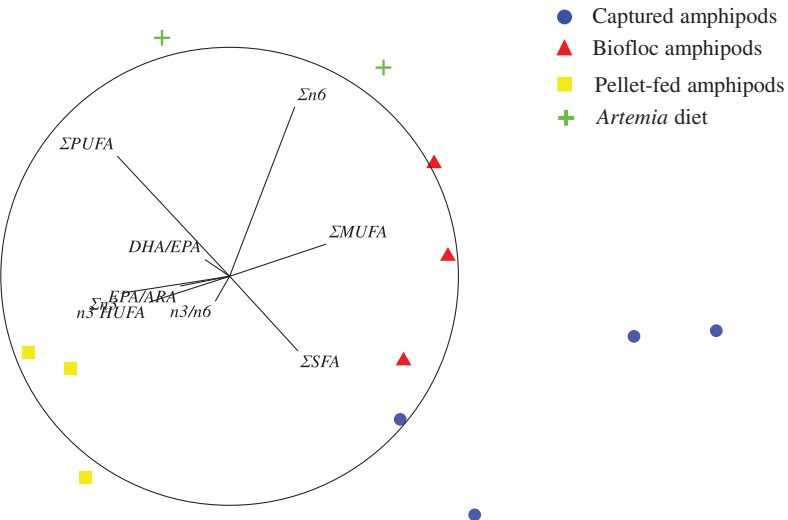

**Figure 2 Non-metric Multidimensional Scaling (2D projection) of indices describing the fatty acid composition of amphipods *P. hawaiensis* of different source and enriched *Artemia*.**

SGR) and in combination with *Artemia* (mixed diet: 11 ± 1 mg day$^{-1}$; 0.67 ± 0.43% SGR), significantly increased the growth rate of juvenile seahorses compared to *Artemia* alone (0.057 ± 0.002 mg day$^{-1}$; 0.04 ± 0.18% SGR; Fig. 3). Moreover, results showed that seahorses fed the *Artemia* diet had growth rates statistically indistinguishable from zero. The GLMM significantly improved (*L.ratio* = 497.5; *p* < 0.001) by adding an exponential variance structure of the form:

$$\sigma^2 = RSE^2 \times exp^{(2\,\delta_i\,Day)}$$

where RSE is the residual standard error, $\delta$ is the parameter for each *i* level of diet. This allowed for the variance associated to each treatment at each experimental day to be

**Table 3 Results of a permutational MANOVA applied on several indices describing the fatty acid composition of amphipods *P. hawaiensis* of different source and enriched *Artemia*.**

| Source of variation | df | SS | MS | *pseudo-F* | *p* | Unique permutations |
|---|---|---|---|---|---|---|
| Food source | 3 | 6777.1 | 2259 | 8.4 | <0.001 | 9,586 |
| Residual | 8 | 2148.7 | 268.6 | | | |
| Total | 11 | 8925.8 | | | | |
| **Post-hoc comparisons** | | | | *pseudo-t* | *p* | Unique permutations |
| Wild amphipods *vs* Biofloc amphipods | | | | 1.8 | <0.05 | 35 |
| Wild amphipods *vs* Pellet amphipods | | | | 3.6 | <0.05 | 35 |
| Biofloc amphipods *vs* Pellet amphipods | | | | 4.2 | 0.09 | 10 |
| Artemia *vs* Biofloc amphipods | | | | 2.6 | 0.09 | 10 |
| Artemia *vs* Pellet amphipods | | | | 3.4 | 0.09 | 10 |
| Artemia *vs* Wild amphipods | | | | 2.5 | 0.07 | 15 |

Note:
df: degrees of freedom; SS multivariate sums of squares; multivariate mean squares; *pseudo-F*, *pseudo-t* and *p*: F and t values obtained through permutations of the reduced model and the *p* values associated; number of unique permutations used to obtain each *pseudo-F* and *pseudo-t* value.

**Table 4 Results of *t*-tests comparing the intercepts and slopes of three lineal regressions on the changes in wet weight (g) of *H. erectus* as a function of time (days) when fed the experimental diets.**

| | Intercept | | Slope | |
|---|---|---|---|---|
| | Amphipod | Mixed | Amphipod | Mixed |
| Mixed | 0.47 ns | – | 1.08 ns | – |
| Artemia | 0.16 ns | 0.62 ns | 8.76 *** | 6.49*** |

Note:
Amphipod: amphipod diet; *Artemia*: *Artemia* diet; Mixed: mixed diet, ns: non-significant; ***$p < 0.001$ (see text for details on the GLMM adjusted to the data).

estimated (Table 5). These results indicate that inclusion of amphipods in the diet made seahorse weight more variable and dispersion increased with experimental days.

Table 6 shows the FA composition of seahorses at the end of the dietary treatment. PCoA on FA composition of seahorse tissue showed that 80.5% of total variation in the data was contained in the first and second principal coordinates. Ordination clearly separated samples from seahorses fed diets that included amphipods to the right-hand side, from those fed *Artemia* alone to the left-hand side of the map (Fig. 4). Eigenanalysis showed that the former were samples with high content of eicosatrienoic (C20:3n3), lauric (C12:0) and myristic (C14:0) and to a lesser extent of eicosenoic (C20:1n9), alpha-linolenic (C18:3n3), eicosapentaenoic (C20:5n3), tridecylic (C13:0), palmitoleic (C16:1) and pentadecylic (C15:0) acids. These samples, however, were low in linoleic (C18:2n6c), gamma-linoleic (C18:3n6), stearic (C18:0), lignoceric (C24:0) and behenic (C22:0) acids. The opposite was true for seahorses fed with *Artemia*. The second coordinate separated samples on the top of the map (mostly from the amphipod diet) with higher contents of oleic acid (C18:1n9c/t), whereas those at the bottom (mostly from the mixed diet) were high in dihomo-gamma-linoleic (C20:3n6), arachidonic (C20:4n6) and docosahexaenoic (C22:6n3) acids (Fig. 4; see Table S3 for details on the contribution of each descriptor to the linear combinations of the first four principal coordinates).

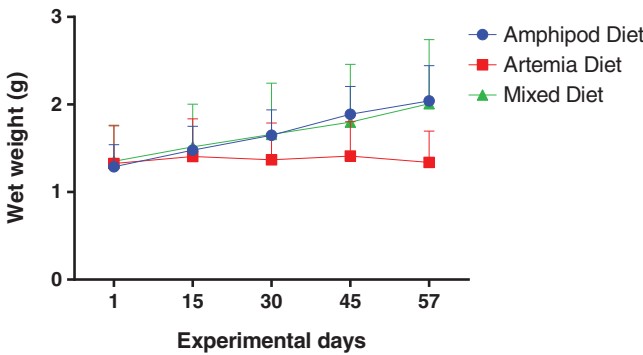

**Figure 3 Wet weight (g) of *H. erectus* fed captured amphipods (amphipod diet), enriched *Artemia* (*Artemia* diet), and a mixture (1:1) of the amphipod and the *Artemia* diets (mixed diet).** Error bars show ± one standard deviation of the mean.

**Table 5 Variance estimates ($\sigma^2$) associated to mean biomass of *H. erectus* fed with three diets (captured amphipods, enriched *Artemia* and a mixed (1:1) diet) on days 0, 15, 30, 45 and 57 of the experiment.**

|  | Diet | | |
|---|---|---|---|
| **Days** | **Amphipod** | **Mixed** | ***Artemia*** |
| 0 | 0.121 | 0.121 | 0.121 |
| 15 | 0.144 | 0.157 | 0.107 |
| 30 | 0.170 | 0.202 | 0.093 |
| 45 | 0.201 | 0.261 | 0.082 |
| 57 | 0.231 | 0.320 | 0.074 |

Note:
Estimates were obtained with a generalized least-square procedure through restricted maximum likelihood and included a variance exponential structure.

Results of the MANOVA confirmed significant differences in FA content associated to seahorse diet (*pseudo-F* = 15.0; $p < 0.001$; 9,920 unique permutations, Table 7). No significant differences were found between subsamples of seahorse tissue (*pseudo-F* = 1.19; $p = 0.28$; 9,907 unique permutations), indicating that the method for the determination of FA content was highly consistent. Paired comparisons of centroids showed that the diet based on *Artemia* resulted in seahorses with a significantly different FA content compared to those fed with either the amphipod or mixed diets (*pseudo-F* = 6.32 and 3.67; both $p < 0.01$; 461 and 462 unique permutations, respectively). Statistical differences were also found between the two diets containing amphipods (*pseudo-F* = 1.94; $p < 0.05$; 461 and 462 unique permutations, Table 7).

The nMDS applied on indices describing the FA profiles also successfully separated tissue samples from seahorses fed with different diets (3D Stress = 0.04; Fig. 5). The permutational MANOVA showed that FA profiles differed significantly depending on diet (*pseudo-F* = 7.63; $p < 0.001$; 9,913 unique permutations; Table 8), whereas variation amongst individual seahorses subjected to the same diet were not larger than those expected by chance (*pseudo-F* = 1.26; $p = 0.24$; 9,914 unique permutations). Here again, the

**Table 6 Fatty acid composition (% of total FAs) of *H. erectus* fed captured amphipods (amphipod diet), enriched *Artemia* (*Artemia* diet), and a mixed diet (1:1; mixed diet) throughout a 57-day experiment.**

| FAs | Amphipod diet | *Artemia* diet | Mixed diet |
|---|---|---|---|
| C12:0 | 0.25 ± 0.06 | 0.09 ± 0.15 | 0.2 ± 0.1 |
| C13:0 | 0.08 ± 0.02 | 0.05 ± 0.01 | 0.1 ± 0 |
| C14:0 | 4.49 ± 0.9 | 1.6 ± 0.22 | 3.6 ± 1.1 |
| C14:1 | 0.03 ± 0.01 | 0.09 ± 0.29 | 0.01 ± 0.01 |
| C15:0 | 0.88 ± 0.15 | 0.59 ± 0.06 | 0.78 ± 0.14 |
| C15:1 | 0.03 ± 0.02 | 0.03 ± 0.03 | 0.02 ± 0.02 |
| C16:0 | 24.05 ± 3.12 | 19.52 ± 2.24 | 24.84 ± 4.16 |
| C16:1 | 8.31 ± 2.2 | 4.72 ± 0.43 | 6.2 ± 2.44 |
| C17:0 | 1.57 ± 0.56 | 1.78 ± 0.27 | 1.5 ± 0.47 |
| C17:1 | 0.53 ± 0.24 | 1.09 ± 0.25 | 0.61 ± 0.38 |
| C18:0 | 12.42 ± 3.93 | 17.72 ± 1.81 | 15.23 ± 2.58 |
| C18:1n9c/t | 20.37 ± 2.96 | 20.28 ± 2.95 | 16.24 ± 4.48 |
| C18:2n6c | 3.43 ± 0.24 | 15 ± 1.09 | 6.21 ± 2.36 |
| C18:3n6 | 0.34 ± 0.08 | 0.58 ± 0.15 | 0.43 ± 0.14 |
| C18:3n3 | 1.01 ± 0.22 | 0.61 ± 0.15 | 0.77 ± 0.17 |
| C20:0 | 0.48 ± 0.09 | 0.56 ± 0.09 | 0.41 ± 0.16 |
| C20:1n9 | 1.43 ± 0.24 | 0.8 ± 0.08 | 1.22 ± 0.28 |
| C20:2 | 0.51 ± 0.06 | 0.35 ± 0.1 | 0.44 ± 0.1 |
| C20:3n6 | 0.35 ± 0.06 | 0.32 ± 0.05 | 0.34 ± 0.07 |
| C21:0 | 0.14 ± 0.02 | 0.17 ± 0.09 | 0.09 ± 0.05 |
| C20:4n6 | 7.48 ± 2.24 | 5.53 ± 1.39 | 8.39 ± 2.33 |
| C20:3n3 | 0.3 ± 0.05 | 0.07 ± 0.04 | 0.22 ± 0.06 |
| C20:5n3 | 4.61 ± 1.68 | 2.39 ± 0.85 | 4.08 ± 1.28 |
| C22:0 | 0.45 ± 0.06 | 0.82 ± 0.11 | 0.49 ± 0.16 |
| C22:1n9 | 0.31 ± 0.05 | 0.3 ± 0.09 | 0.35 ± 0.26 |
| C22:2 | 0.18 ± 0.1 | 0.08 ± 0.06 | 0.15 ± 0.11 |
| C23:0 | 0.09 ± 0.03 | 0.14 ± 0.05 | 0.11 ± 0.06 |
| C24:0 | 0.28 ± 0.05 | 0.45 ± 0.08 | 0.3 ± 0.12 |
| C22:6n3 | 5.6 ± 2.46 | 4.29 ± 1.92 | 6.65 ± 2.8 |
| ΣSFA | 45.18 ± 3.11 | 43.5 ± 4.22 | 47.66 ± 5.62 |
| ΣMUFA | 31.02 ± 4.91 | 27.31 ± 2.95 | 24.67 ± 5.02 |
| ΣPUFA | 23.81 ± 6.41 | 29.21 ± 5.33 | 27.67 ± 7.43 |
| n3 HUFA | 10.51 ± 3.9 | 6.75 ± 2.75 | 10.96 ± 3.61 |
| Σn3 | 11.51 ± 4 | 7.35 ± 2.87 | 11.72 ± 3.64 |
| Σn6 | 11.61 ± 2.43 | 21.42 ± 2.47 | 15.36 ± 4.19 |
| n3/n6 | 0.97 ± 0.17 | 0.33 ± 0.1 | 0.76 ± 0.15 |
| DHA/EPA | 1.22 ± 0.32 | 1.75 ± 0.21 | 1.66 ± 0.59 |
| EPA/ARA | 0.62 ± 0.15 | 0.42 ± 0.08 | 0.5 ± 0.16 |

**Note:**
Data show ± one standard deviation of the mean.

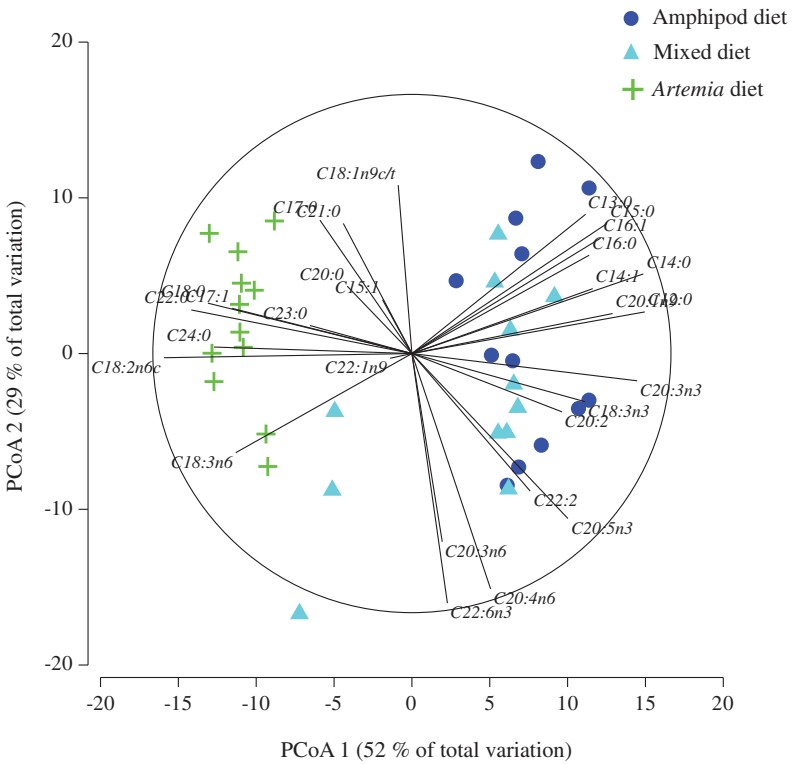

**Figure 4 Principal coordinate analysis of fatty acid composition in tissue samples of *H. erectus* fed captured amphipods (amphipod diet), enriched *Artemia* (*Artemia* diet), and mixed (1:1) diet.**

**Table 7 Results of a permutational MANOVA applied on the fatty acid composition in samples of *H. erectus* fed captured amphipods (amphipod diet), enriched *Artemia* (Artemia diet), and mixed (1:1) diet.**

| Source of variation | df | SS | MS | *pseudo-F* | *p* | Unique permutations |
|---|---|---|---|---|---|---|
| Diet | 2 | 2607.2 | 1303.6 | 15.0 | <0.001 | 9920 |
| Individual (Diet) | 15 | 1301.9 | 86.8 | 1.2 | 0.28 | 9907 |
| Residual | 18 | 1310.4 | 72.8 | | | |
| Total | 35 | 5219.5 | | | | |
| **Post-hoc comparisons** | | | | *pseudo-t* | *p* | Unique permutations |
| Amphipod *vs* Mixed diet | | | | 1.9 | <0.05 | 462 |
| Amphipod *vs* *Artemia* diet | | | | 6.3 | <0.01 | 461 |
| *Artemia vs* Mixed diet | | | | 3.7 | <0.01 | 462 |

**Note:**
df, degrees of freedom; SS multivariate sums of squares; multivariate mean squares; *pseudo-F*, *pseudo-t* and *p*: *F* and *t* values obtained through permutations of the reduced model and the *p* values associated; number of unique permutations used to obtain each *pseudo-F* and *pseudo-t* value.

FA profile of seahorses fed with *Artemia* was statistically different from the amphipod and mixed diets (*pseudo-F* = 4.23 and 2.26; both $p < 0.01$; both 462 unique permutations, respectively), but these two were not statistically distinguishable (*pseudo-F* = 1.81; $p = 0.06$; 462 unique permutations). The 3D configuration projected on two dimensions showed that samples from the amphipod and mixed diets had high Σn3, n3HUFA, EPA/ARA and
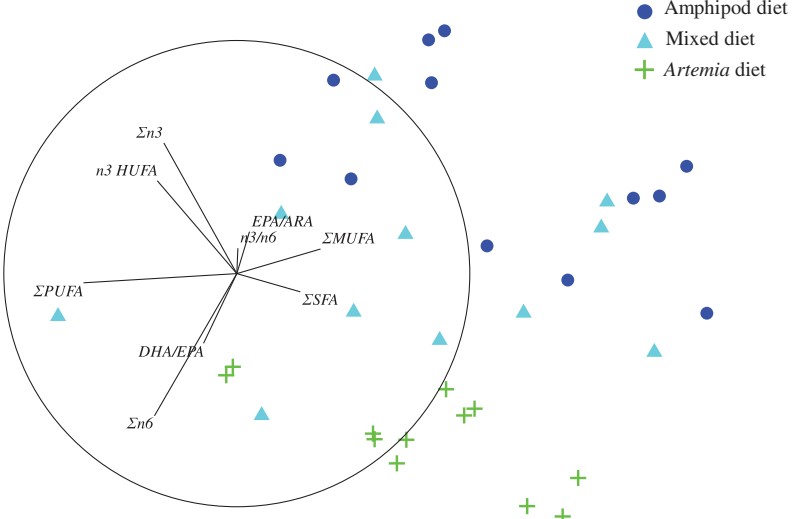

**Figure 5 Non-metric Multidimensional Scaling (2D projection) of indices describing the fatty acid composition in tissue samples of seahorses *H. erectus* subjected to different dietary treatments.** Amphipod diet: captured amphipods; *Artemia* diet: enriched *Artemia*; mixed diet: a 1:1 mixture of the amphipod and Artemia diets.

**Table 8 Results of a permutational MANOVA applied on indices describing the fatty acid composition of *H. erectus* fed captured amphipods (amphipod diet), enriched *Artemia* (Artemia diet), and mixed (1:1) diet.**

| Source of variation | df | SS | MS | *pseudo-F* | *p* | Unique permutations |
|---|---|---|---|---|---|---|
| Diet | 2 | 5799.7 | 2899.8 | 7.6 | <0.001 | 9,913 |
| Individual (Diet) | 15 | 5701.6 | 380.1 | 1.3 | 0.24 | 9,914 |
| Residual | 18 | 5419.7 | 301.1 | | | |
| Total | 35 | 16,921 | | | | |
| **Post-hoc comparisons** | | | | *pseudo-t* | *p* | Unique permutations |
| Amphipod *vs* Mixed diet | | | | 1.8 | 0.06 | 462 |
| Amphipod *vs Artemia* diet | | | | 4.4 | <0.01 | 462 |
| *Artemia vs* Mixed diet | | | | 2.3 | <0.01 | 462 |

**Note:**
df, degrees of freedom; SS multivariate sums of squares; multivariate mean squares; *pseudo-F*, *pseudo-t* and *p*: *F* and *t* values obtained through permutations of the reduced model and the *p* values associated; number of unique permutations used to obtain each *pseudo-F* and *pseudo-t* value.

ΣMUFA values, whereas those from *Artemia* had high DHA/EPA and Σn6 (Fig. 5). The n3/n6 ratio was higher in the former than in the latter. The amphipod diet resulted in slightly higher ΣMUFA, whereas the mixed diet had higher ΣPUFA (Fig.5); but these differences can only be considered marginally significant (see Table S4 for details on the multiple correlation coefficients of descriptors with the first three ordinal axes).

# DISCUSSION

## Nutritional value of *P. hawaiensis* from different production sources

Results in the present study revealed that captured *P. hawaiensis* contains high levels of lipids (20.4 ± 0.8% of dw), with a FA profile suitable for feeding *H. erectus* juveniles,

namely rich in the EFAs ARA, EPA and DHA. Interestingly, amphipod production based both on the use of a large biofloc system and a small in-door tank with commercial pellets tended to increase the lipid content (26.7 ± 1.3% and 25.5 ± 3.5%, respectively). This was true despite the relatively low lipid content of the commercial feed used (80 g lipid kg$^{-1}$) and the typically low content of bioflocs (1.6–8.3% lipids) (*Ahmad et al., 2017*; *Magaña-Gallegos et al., 2018*; *Sgnaulin et al., 2018*). There is a limited number of studies analyzing lipid content in the context of aquaculture to compare with, but these lipid levels are slightly higher than those found in amphipods harvested from an off-shore aquaculture farm (13%) (*Fernandez-Gonzalez et al., 2018*), shrimp biofloc ponds (4.7–6.3%), rivers (7.5–13%) (*Kolanowski, Stolyhwo & Grabowski, 2007*), different marine areas (5.1–19.15%) (*Baeza-Rojano, Hachero-Cruzado & Guerra-García, 2014*; *Jiménez-Prada et al., 2018*), as well as in other commonly used live food organisms, such as mysids (6.7–8.0%) (*Planas et al., 2020*) and copepods (11.3–12.4%) (*Wang et al., 2014*). The optimal dietary lipid levels for seahorses are still unknown, but researchers agree that beyond lipid content, a well-balance FA fraction plays a pivotal role in seahorse nutrition (*Faleiro & Narciso, 2010*; *Segade et al., 2016*; *Planas et al., 2020*).

Amphipods from all three sources (captured, biofloc and pellet-fed amphipods) showed valuable levels of PUFAs (26.38–41%), almost as high as SFAs (39.6–41.7%). These levels are similar to those reported for the marine gammarid *Hyalella media* collected from the same coastal area (*Baeza-Rojano et al., 2013b*) and others from the strait of Gibraltar (*Baeza-Rojano, Hachero-Cruzado & Guerra-García, 2014*); but lower than highly nutritious organisms such as copepods (*Zhang, Lin & Liu, 2015*), mysids (*Schlechtriem, Arts & Johannsson, 2008*; *Herrera et al., 2011*) and long-time enriched *Artemia* (*Planas et al., 2017*), particularly regarding DHA and n3/n6 ratio. In contrast to enriched *Artemia*, that showed a FA profile rich in Σn6 and linoleic acid typical of freshwater organisms (*Sargent et al., 1999*), amphipods were characterized by high contents in SFAs and MUFAs. Amphipods did not dramatically differ in terms of the presence of nutritionally relevant FAs, such as ARA, EPA and DHA, compared to *Artemia*. However, both PCoA and nMDS ordinations clearly separated amphipods from *Artemia*, suggesting that amphipods used herein could present a more "marine profile", similar to that in studies of amphipods from littoral areas (*Woods, 2009*; *Jiménez-Prada et al., 2018*; *Alberts-Hubatsch, Slater & Beermann, 2019*). This was especially true for pellet-fed amphipods, which showed the highest values of Σn3, n3 HUFAs, n3/n6, DHA, DHA/EPA and EPA/ARA. DHA, along with ARA and EPA, are important components of cellular membranes and precursors of bioactive molecules involved in essential metabolic and physiological processes (*Tocher, 2003*). These FAs are considered essential for marine fish and shrimp nutrition, as they have limited enzymatic capacity to synthesize them *de novo* (*Tocher et al., 2003*). Required absolute values for these nutrients are species-specific, but they are generally needed at levels around 1% dry weight of diet (*National Research Council (NRC), 2011*). Besides providing sufficient amounts of these FAs to meet requirements, it is also important to optimize their relative proportions, as their essential functions can be influenced by the presence and relative amounts of other FAs (*Izquierdo & Koven, 2011*). For example, in Atlantic salmon, addition of EPA to a DHA rich diet enhanced DHA tissue

retention and fish growth, whereas addition of ARA did not enhance growth but reduced DHA retention (*Glencross et al., 2014*). Although captured amphipods in the present study showed higher ARA contents (Figs. 1 and 2), DHA was highest in both biofloc and pellet-fed amphipods, suggesting a better FA balance in these cultured amphipods. Absolute FA amounts were not quantified in the present study but, with ARA, EPA and DHA contents ranging from 2.9 to 9.1% of total FAs, *P. hawaiensis* could fulfill such dietary requirements (*National Research Council (NRC), 2011*; *Zhang, Lin & Liu, 2015*).

The improved FA profile of pellet-fed amphipods, and to a lesser extent of biofloc amphipods, was not surprising, given that the commercial shrimp feed used is expected to meet the nutritional requirements of penaeid shrimps (*Martinez-Cordova, Campaña Torres & Porchas-Cornejo, 2003*). The main ingredient of the feed is fishmeal (*Chávez-Sánchez, 1993*), which may contain ideal sources of PUFAs and amino acids for aquafeeds (*Vargas-Abúndez et al., 2019*). Whilst both biofloc and pellet-fed amphipods were fed the commercial shrimp diet, it is likely that the biofloc culture provided an additional source of food for the amphipods, hence explaining the slight differences in FA composition between biofloc and pellet-fed amphipods. *P. hawaiensis* is a detritivorous species and, as an opportunistic grazer, it is expected to feed efficiently on different floc particles. Amphipod gut content was not analyzed but amphipods introduced into the biofloc system changed markedly in color compared to pellet-fed amphipods, turning from almost translucence to a dark brown body with a green belly after a few days (personal observations), possibly as a result of microalgae and biofloc ingestion.

As previously reported for a similar biofloc setup (*Magaña-Gallegos et al., 2018*), the FA composition of the biofloc particles in the present study was probably low in EPA and DHA. However, the actual nutritional value of bioflocs can vary according to a number of factors, including biofloc particle size, carbon source, biofloc maturation, floc density, food preference by cultured animals and their ability to ingest and digest the different biofloc particles (*Ahmad et al., 2017*; *Magaña-Gallegos et al., 2018*; *Promthale et al., 2021*). Results herein are insufficient to ascertain specific causes, but indicate that resulting biofloc amphipods present a FA profile highly suitable for applications in marine aquaculture, *i.e.*, with significant amounts of ARA, EPA and DHA and better n3/n6, DHA/EPA, EPA/ARA ratios compared to enriched *Artemia*. Further research should assess other nutrients such as proteins and their amino acid constituents, as they also play foremost important roles in fish growth and development (*D'Abramo, 2019*).

## Effect of amphipods on the seahorse growth and fatty acid profile

The use of captured amphipods substantially enhanced the growth of juvenile seahorses when used both solely (amphipod diet) or in combination with enriched *Artemia* (mixed diet) (Fig. 3). Whilst all individuals in all dietary treatments survived throughout the 57-day trial, the use of enriched *Artemia* as the only food source did not promote seahorse growth, since no significant change in the wet weight in animals in this treatment could be demonstrated. Results similar to these have been previously reported in other *H. guttulatus* adults (*Palma et al., 2008*). It is well documented that *Artemia* is not an adequate prey for many seahorse species due to nutritional deficiencies, poor digestibility and absorption

(*Payne & Rippingale, 2000*; *Blanco & Planas, 2015*; *Randazzo et al., 2018*). Despite its limitations, *Artemia* is largely used given its extensive availability (*Sorgeloos, Dhert & Candreva, 2001*; *Bengtson, 2003*; *Olivotto et al., 2008*; *Del Vecchio et al., 2019*; *Southgate, 2019*; *Planas et al., 2020*).

In contrast to *Artemia*, amphipods are a natural prey for seahorses and other marine organisms (*Manning, Foster & Vincent, 2019*). Several authors have found food preferences in seahorses over natural preys, such as mysids and copepods (*Buen-Ursua et al., 2015*, *Blanco & Planas, 2015*), although depending on the stage of development. Regarding amphipods, a previous study by the authors (*Vargas-Abúndez, Simões & Mascaró, 2018*), found very different ingestion rates in *H. erectus* juveniles (7.7–96 mm standard length) when comparing amphipods and *Artemia* diets. With frozen amphipods (*E. pectenicrus*), seahorses ingested 4.1 ± 1.7% of its wet body weight within 12 min, whereas with live *Artemia* seahorses ingested equivalent biomass only after 90 minutes. In the present study, ingestion rates were not measured, but based on the previous one, it is likely that fish ingested a higher amphipod biomass compared to that of *Artemia*, thereby partially explaining the increased growth in the two diets that contained amphipods. These results are encouraging and consistent with previous observations on other marine organisms kept under controlled conditions. In a trial with *Octopus maya*, live marine gammarids induced a higher growth rate (6.9 ± 0.2% day$^{-1}$) and survival (92.2 ± 6.8%) compared to adult *Artemia* (4.8 ± 0.2% day$^{-1}$ growth and 74.5 ± 23.8% survival) and freshwater gammarids (5.0 ± 0.3% day$^{-1}$ growth and 41.2 ± 21.2% survival) (*Baeza-Rojano et al., 2013b*). Both gammarids and caprellids have been recognized as nutritionally adequate prey for cuttlefish hatchlings (*Baeza-Rojano et al., 2010*). However, gammarids induce a better feeding response and consequently growth and survival in cuttlefish, compared to caprellids, due to differences in behavior (*Baeza-Rojano et al., 2010*). In Atlantic salmon and Atlantic halibut, amphipod meal successfully replaced up to 40% of fish meal in compound diets, with no negative effects on feed conversion ratio, dry matter digestibility, protein digestibility and muscle composition (*Suontama et al., 2007*).

The improved growth of *H. erectus* fed either the amphipod or the mixed diet could also be related to the nutritional value of the diets, particularly with regard to FAs. The nutritional requirements of seahorses are still unknown, but high levels of PUFAs, particularly of the LC-PUFAs ARA, EPA and DHA, seems to be determinant for seahorse growth and survival (*Faleiro & Narciso, 2010*; *National Research Council (NRC), 2011*). In the wild, seahorses mostly consume small crustaceans such as copepods, mysid shrimps and amphipods (*Teixeira, Musick & Musik, 2001*; *Manning, Foster & Vincent, 2019*), known to be rich sources of LC-PUFAs (*Woods, 2009*; *Guerra-García et al., 2014*; *Alberts-Hubatsch, Slater & Beermann, 2019*).

In the present study, both PCoA and nMDS ordination methods applied on their respective FA profiles (Figs. 4 and 5) clearly separated samples of seahorses fed either amphipod diets from those fed *Artemia*, and this separation was statistically distinguished from random noise. *Artemia* was enriched with PUFAs (Super Selco®) and thus resulted in higher PUFA percentages with respect to both amphipod diets. However, the higher PUFA content was mainly the result of increased linoleic acid (C18:2n6c), whereas

nutritionally relevant markers such as, ARA, EPA and DHA were found in similar percentages in all three diets. Linoleic acid is characteristic of terrestrial plants and consequently of non-marine organisms such as *Artemia*, especially when fed ingredients of terrestrial origin (*Balachandar & Rajaram, 2019*). Linoleic acid is considered an EFA for fresh-water fish, but not for marine fish (*National Research Council (NRC), 2011*). In the present study, *Artemia* was raised with wheat bran, which could have increased the abundance of this FA. Interestingly, the higher PUFA content in *Artemia* did not translate into increased levels of PUFAs in the seahorses. On the contrary, despite similar ARA, EPA and DHA compositions amongst all three diets, seahorses fed enriched *Artemia* showed lower percentages of these FAs. This trend became more clearly evidenced by the nMDS projection applied to indices describing the FA profile, where samples of seahorses fed either amphipod diets showed closer association to vectors representing Σn3, n3 HUFAs, EPA/ARA and ΣMUFAs, whereas those from *Artemia* were closer to Σn6 and DHA/EPA (Fig. 5). In fact, the higher n6 percentage detected in the *Artemia* diet was reflected in the seahorse n6 percentage and, consequently, in a substantial reduction in the n3/n6 ratio amongst seahorses fed with this diet. Marine fish have higher requirements of n3 FAs than of n6, and, as stated earlier, optimal ratios tend to be high due to competitive interactions in FA biosynthesis (*Faleiro & Narciso, 2010*; *National Research Council (NRC), 2011*). n3/n6 ratios of about 2.5–3.5 are common in natural prey ingested by seahorses (*Zhang, Lin & Liu, 2015*; *Segade et al., 2016*; *Planas et al., 2020*). In egg, newborn and juvenile seahorses, this ratio ranges from 1 to 16, being generally higher in earlier developmental stages (*Saavedra et al., 2014*; *Segade et al., 2016*; *Planas et al., 2020*). In newborn *H. erectus*, improved growth with an n3/n6 ratio of 2.5 was achieved with the use of calanoid copepods collected from fish ponds (*Zhang, Lin & Liu, 2015*). Thus, the n3/n6 ratio observed in the present study when using the amphipod diet (0.97 ± 0.17) was comparatively low but highly satisfactory for feeding late juveniles of *H. erectus*.

A third element that could explain the better performance of the amphipod diets is digestibility. Poor digestibility due to limited enzymatic capacity is a common issue in newborn and early stage seahorses (*Blanco & Planas, 2015*; *Novelli et al., 2016*; *Ofelio et al., 2018*). However, it was suggested that adult seahorses can display differences in the digestibility of zooplanktonic organisms (*Corse et al., 2015*). The digestibility of amphipods has only been evaluated in Atlantic salmon and Atlantic halibut compound diets, and it was comparable to the excellent digestibility of fish meal and krill meal (*Suontama et al., 2007*). That brings the possibility that amphipods, as natural prey of *H. erectus*, could be more efficiently digested and absorbed compared to *Artemia*. It may explain why diets containing amphipods outperformed the *Artemia* diet despite small differences in key nutrients (ARA, EPA and DHA). It remains unclear why seahorses fed exclusively *Artemia* did not display any growth after 2 months. More information on other nutrients such as energy and protein content, as well as amino acid profiles, which are essential for fish growth, is required to properly address this issue.

In the present study, neither pellet nor biofloc amphipods were used in the seahorse feeding trials, but given their FA profile, it can be expected similar or even enhanced fish performances compared to captured amphipods. Further research is required to confirm

the suitability of using cultured amphipods. The use of biofloc for amphipod production represents a more sustainable and cost-effective new technology (*Ahmad et al., 2017*), which could be easily scalable for the commercial production of valuable feed for marine fish, well beyond seahorses.

## CONCLUSIONS

Both captured and cultured *P. hawaiensis* showed adequate levels of lipids and n3 fatty acids. *H. erectus* fed with captured *P. hawaiensis*, either alone or in combination with enriched *Artemia*, improved seahorse growth and fatty acid profiles in terms of ARA, EPA, DHA, Σn3 percentages, as well as n3/n6 ratio. The present research supports the potential use of amphipods as an alternative prey for feeding seahorses. Further research addressing the nutritional value of other important nutrients, such as amino acids and microelements is required for a comprehensive understanding of the amphipod nutritional value.

## ACKNOWLEDGEMENTS

The authors are grateful to Eduardo Cruz-Hernandez for conducting the seahorse feeding trial, collecting and processing the corresponding data, and to Marcela Yamileth Lopez Noriega for collecting and processing amphipod samples for fatty acid analysis.
M. Sc. Iveth Gabriela Palomino Albarrán and Patricia M. Balam Uc supplied live *Artemia*.

### Funding

This study was carried out with the financial support of projects Programa Apoyo a Proyectos de Investigación e Innovación Tecnológica [PAPIIT IN212012, IN219816 and IN219319], The Academia Mexicana de la Ciencia granted a scholarship to Marcela Yamileth Lopez Noriega for assisting the investigation. The funders had no role in study design, data collection and analysis, decision to publish, or preparation of the manuscript.

### Grant Disclosures

The following grant information was disclosed by the authors:
Programa Apoyo a Proyectos de Investigación e Innovación Tecnológica: PAPIIT IN212012, IN219816 and IN219319.
The Academia Mexicana de la Ciencia.

### Competing Interests

The authors declare that they have no competing interests.

### Author Contributions

- Jorge Arturo Vargas–Abúndez analyzed the data, prepared figures and/or tables, authored or reviewed drafts of the paper, and approved the final draft.
- Gemma Leticia Martínez–Moreno conceived and designed the experiments, performed the experiments, authored or reviewed drafts of the paper, and approved the final draft.

# PeerJ

- Nuno Simões conceived and designed the experiments, authored or reviewed drafts of the paper, and approved the final draft.
- Elsa Noreña–Barroso analyzed the data, authored or reviewed drafts of the paper, and approved the final draft.
- Maite Mascaró conceived and designed the experiments, analyzed the data, prepared figures and/or tables, authored or reviewed drafts of the paper, and approved the final draft.

## Animal Ethics

The following information was supplied relating to ethical approvals (*i.e.*, approving body and any reference numbers):

The present study was carried out under a permit by Mexican Ministry of the Environment and Natural Resources (SEMARNAT) No. SGPA/DGVS/12741/13 and strictly followed institutional protocols for the maintenance, manipulation, and sacrifice of the experimental animals according to certified criteria established by the Guide for the Care and Use of Experimental Animals in Research and Teaching of the Faculty of Superior Studies-Cuautitlán at Universidad Nacional Autónoma de México.

## Field Study Permissions

The following information was supplied relating to field study approvals (*i.e.*, approving body and any reference numbers):

Collection of seahorses was performed under a permit granted by Mexican Ministry of the Environment and Natural Resources (SEMARNAT; No. SGPA/DGVS/12741/13).

## Data Availability

All raw data are available in the Supplemental Files.

## Supplemental Information

Supplemental information for this article can be found online at http://dx.doi.org/10.7717/peerj.12288#supplemental-information.

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
