# Peer review of "Marine amphipods (Parhyale hawaiensis) as an alternative feed for the lined seahorse (Hippocampus erectus, Perri 1810): nutritional value and feeding trial"

_PeerJ, doi:10.7717/peerj.12288_

## Round 0.1 · original submission · Major Revisions

I now have three reviews from experts in aquaculture. Two of the reviewers raise substantive concerns about the methods used in this study that could be grounds for declining to publish this manuscript, but I am offering the authors an opportunity to address them in a major revision. Note particularly the concerns of Reviewer 2 regarding the fatty acid data and statistical methods. If the authors choose to submit a revision, I will need very clear point-by-point responses to the reviewer comments for their reconsideration.

Reviewer 1 ·

Basic reporting

no comments

Experimental design

The Materials & Methods section needs the seawater temperature of the experiment.

Validity of the findings

no comments

Additional comments

The present MS reported the nutritional value of the marine Amphipoda Paryhale hawaiensis in the wild captured and cultured one, and provide these amphipods to juveniles of seahorse compared with Artemia. This study presented well that the usage of Paryhale hawaiensis is one of the promising feeding items to seahorse aquaculture. While the present form of MS needs several minor revisions, the results would be worthy of publishing in one of the international scientific journals.

My comments for improving the MS are,

1) The introduction needs more detail, especially a clear explanation on what is the progress of the authors' previous studies;

Vargas-Abundez et al. (2018) Feeding the lined seahorse Hippocampus erectus with frozen amphipods. Aquaculture 491, 82–85.
Vargas-Abundez et al. (2021) Marine amphipods as a new live prey for ornamental aquaculture: exploring the potential of Parhyale hawaiensis and Elasmopus pectenicrus. PeerJ 9, e10840.

2) The introduction needs a more detailed explanation of the characteristics of the life history and ecology of Paryhale hawaiensis based on numerical data.

3) What is a plastic label fastened to the neck of the seahorse (Line 213)? If the authors could prepare photos on the label, readers could understand the rearing method.

4) Figure 3 was based on the estimated regression. I think that Figure on the average and S.D. would help readers to understand the validity of estimation.

5)The discussion would need a more detailed explanation of what the purpose of seahorse aquaculture is and why the usage of amphipod Paryhale hawaiensis is suitable for this aquaculture purpose.

Reviewer 2 ·

Basic reporting

I found the research topic interesting. The idea of culturing a food item in a biofloc system is attractive. It is scalable and would allow for the commercial production of valuable feed for marine fish. This could go well beyond seahorses.

Experimental design

The study design has some serious issues and I question a few of the key aspects of the design which limits the study publication potential. Your diet study and your biofloc culture appear to be disconnected but joined into this manuscript.

Feel free to correct me, unfortunately I feel the following issues warrants rejecting the manuscript:

The diet study design:

1. Where is the biofloc, pellet grown feed in the diet study?
LINES 212-217 i) amphipod diet: 100% captured amphipods; ii) Artemia diet: 100% enriched Artemia; iii) mixed diet: a 1:1 mix of the amphipod and the Artemia diets.

It is possible that the seahorses do well on 'captured' amphipods but do poorly on cultured amphipods? If so that would invalidate the premise of growing the amphipods for food. Their fatty acid profiles are very different and this would seem important.

2. Your starting weights for the seahorses are not reported correctly. You report a mean and S.D. for your starting weights, but what you should report is a range. Your range was 0.72 --- 2.48 with a mean of 1.3. This starting weight invalidates your study design; unless you can demonstrate that starting weight had no effect on weight gain or SGR. Given that your starting weight is so vastly different and it is likely that your seahorses were at different development states. Clearly, your artemia feed treatment does not provide the nutrition needed as the fish didn't gain weight. Weight gain and SGR are tied to development and age.

3. Your fatty acid raw data shows some issues. It appears that your duplicate samples (mentioned on LINES 251-252) indicate some method issues. It is to be commended to take duplicate samples in a diet study, the purpose is to check analytical precision. In chemistry, it is expected that two subsamples from the same sample would yield nearly identical results. If results are not identical or close to identical it indicates an error in the process. Samples 1-a and 1-b, provide a good example. These do not look like the same starting materials.


1-a 1-b
C22:6n3 8.71 2.37
Σ SFA 42.83 44.39
Σ MUFA 26.00 41.47
Σ PUFA 31.17 14.14

These differences cause this reviewer to believe that you have a process error in your fatty acid prep or sampling.

Validity of the findings

I'm concerned about the above issues confounding your findings.

Additional comments

Great topic, writing needs some work, the manuscript is too long, needs some focus. The issues with experimental design should result in a rejection or a major rewrite and resubmission. I would argue for a rejection.

Reviewer 3 ·

Basic reporting

No comment, except the abstract and introduction length. In my personal opinion abstract is too long (250 words is enough). Concerning the 'introduction': It is well written, clear and concise, focusing from general to precise aim of the paper. however some paragraphs should be summarized (e.g. L83-100): Sometimes there are many references (L79/L84).

Experimental design

The second objective should be better defined and seahorse species should be included in the paper objectives. Ideally some hypothesis linked to objectives would be desired as well.
Methods are well described and statistical analysis is well described. However It would be nice if authors can explain better Fatty Acids determination on seahorses: "Fatty acid determinations were performed on seahorses once at the end of the trial. Duplicate subsamples from each of six individuals fed each of the three diets were randomly taken for this purpose". Authors should clarified if animals were scarified or not, how, if the entire animal was used for analysis or just some tissues. It is unclear for me as well, from the 48 seahorses (16 for each treatment), how many and the selection criteria employed to sample tissues.

Validity of the findings

The study presents the use amphipods as an alternative live prey for seahorse (Hippocampus erectus) aquaculture. The literature justify the research presented as some bottlenecks, not only in seahorse, in general in ornamental aquaculture of marine species exist since many years compromising their own sustainability. Authors described also the lipid composition of targeted amphipod species enlarging the knowledge about nutritional quality of these preys.
Results: Lipids values are expressed in %, but authors should specify in this % is in dry or wet mass. Other authors should express lipids as g/100 g dry weight. It would be easy to compare with values obtained in other seahorse preys such as mysids (Herrera et al. 2010).

I have some concerns on the discussion, which sometimes showed some speculative sentences that I described later on General comments section.
Finally for conclusions, It seems more a summary of ideas. Conclusions should be reformulated with the main demonstrated facts extracted from the paper.
L578-582: This two sentences should be summarized and merged on only one.
L582. Considerations about biofloc technology can be inside the discussion but not as paper conclusion.
The improvements in FA profiles could be linked to statistical differences, which there are not specifically tested on the paper, therefore we can not assume them as a conclusion.

Additional comments

In general the paper is well written and should be published. However I have main thought regarding the study: In general it presents the utility of amphipods as alternative prey for seahorses promoting their growth and showing also their lipids and FA profile. Ideally some reflection about protein content should be included in the discussion, as this information is missed in the paper. High quality lipids (specially HUFA) are important in the early phases of development of any organism, but for growth, proteins are at least very important too.

Specifically there is some comments described as follows:
L211. Seahorse Feeding trial. Seahorses had 10 months when the experiment starts. I am just wondering if during the two additional experimental months any reproduction behaviour was registered. One year in other seahorse species implies mature specimens that when they are fed with high quality food (in this experiment animals were even overfed) involve at least courtship behaviours. In addition there is no data about sex ratio (males:females) and their length measurements (only weight). Normally females can growth faster as normally they are less involved in reproduction (compared to males). It would be nice if authors can add some comments about it.
L438. Authors observed that " ...amphipod introduced into the biofloc system changed markedly in color compared to pellet-fed amphipods." as a result "...microalgae and biofloc ingestion". This statement is merely speculative as they said "amphipod gut content was not analyzed". It would be ideal if authors can add a figure showing amphipod color changes.
L451 and L462. These two paragraphs state the benefits of biofloc technology. Viewing the length of the discussion, I suggest that this information was summarized in few lines always trying to find the relationship with the paper outcomes.
L470. 4.2. Effect of amphipods on the seahorse performance and fatty acid profile. I suggest to change "performance" by "growth" as authors did not assess other parameter except animals growth.
L499. Authors state that seahorses "... likely ingested a higher amphipod biomass compared to that of Artemia, thereby partially explaining the increased growth in the two diets that contained the former." It should be nice to add some results regarding ingestion acceptability. Did the authors observe any difference between preys ingestion?
L557. Statements about digestibility are based on some references that mostly focused on studies carried out during seahorse early developmental stages (Blanco et al., 2015; Novelli et al., 2016; Ofelio et al., 2018) , which it is not the case of experiments developed on the manuscript (seahorse are almost 1 year old). Therefore this issue should be clarified or references deleted. Moreover the rest of paragraph state different hypothesis to explain digestibility issues which there are not tested under this manuscript. Therefore even if it is important to pointed out, text should be summarized.
L555. This paragraph can be deleted or merged summarized somewhere before in the discussion.
L561. I am partially agreed with authors statement about "... monodiets inconvenience to provide essential nutrients". It always depends on the nutritional quality of preys: Wild-capture mysids has been used successfully employed in the reproduction of seahorses (H. reidi) as monodiets (Otero-Ferrer et al. 2016, 2020). On the other hand feeding regimes can be adapted to animal life-stage or culture objective: e.g. broodstock feeding regimes should target high-quality lipid content, instead juveniles and subadults should search high-quality protein content. In this context amphipods seems to be a prey in which its nutritional quality can be easily modulated.

---

## Round 0.2 · Minor Revisions

All three reviewers found the revision to be an improvement, but as you can see, there remain concerns, primarily those of Reviewer 2. Please prepare another revision that addresses the second set of comments of the reviewers. In particular, as Reviewer 2 suggests, please provide additional information in graphs and tables and additional explanation.

Reviewer 1 ·

Basic reporting

The present study assessed the nutritional value of a marine gammarid Paryhale hawaiensis and evaluated through a feeding trial its potential use as a natural prey for the lined seahorse Hippocampus erectus. The authors indicated this species of gammarid possess high levels of valuable lipids and polyunsaturated fatty acids, including the long-chain PUFAs (LC-PUFAs) arachidonic acid (ARA) and so on. The authors’ experiment indicates that gammarid Paryhale hawaiensis is a more suitable prey item for the lined seahorse than Artemia. I understood that the authors responded to all comments and improved the revised MS. I think that the revised MS is suitable for publication.

Experimental design

no comment

Validity of the findings

no comment

Reviewer 2 ·

Basic reporting

The basic reporting of this revision remains mostly unchanged from the original submission. I found the revisions to help the flow of the manuscript.

A few areas that need to be greatly improved if this moves forward.

The graphs:

The PCoA is not interpreted as well as it should be. The vectors and eigenvalues should be reported. The results of the PCoA are not presented with enough information to determine the validity of your data analysis. For example, there are clear outliers in your data set from the raw data (see Experimental Design discussion), because your number of replicates and number of factors are so similar, PCoA might be a better choice here than a PCA, but outliers have large effects on PCoA with small samples. In fact, the use of PCA and PCoA to find outliers is a very useful method of analysis for machine learning. If you do a literature search for PCA/PCoA and fatty acids (FAMEs), you will see that many authors reduce the dataset down to the most important fatty acids. While there is no hard and fast rule on sample size > factors, outliers are real problems with small data sets. Furthermore, from your vectors, you can see some clustering of vectors into groups, it is possible for some values to strongly affect others and therefore masking your data.

You should at a minimum provide some interruption of your data that allows the reader to understand these impacts. I would suggest you reduce the number of factors down greatly, and discuss the vectors (FAMEs). Most marine FAMES papers do not focus on FAMES less than 16.

I'm not exactly sure why you chose to use two different multivariate methods (one on individual FAME values, others on value groupings/ratios). You clearly have a reason for this, perhaps you can improve the literature around multivariate methods by providing some discussion on why PCoA or nMDS was chosen. Why not a PCA? Most applications are using PCAs for FAMEs.

Can you provide some incite here?

I'm not sure how informative your nMDS is compared to the PCoA. These reduction methods are really great data exploration methods but sometimes abstract representations of the data. You should use the most informative information and present that in the body of the manuscript and provide supporting data in your supplement information.

I would suggest you cluster the groupings onto your PCoA graph and show these in one figure.

Explore the R vegan package or Python skbio for better utilities.

Your growth graph should be fit to the model data and not connect the dots. Show the positive and negative errors around your data.

Graph quality could be greatly improved using GraphPad for all your graphs or using R

The Tables:

Add tables to support your multivariate work. Can be in the supplemental.

Your tables 1 and 4 do not say if you are reporting S.E. or S.D. Recall that S.E. is used for reporting data where the sample size changes. If your samples size is the same throughout then report S.D.

Experimental design

The authors deal with the size difference in the rebuttal. However, I still would caution that growth is very impacted by the life stage. Diet studies normally want to see an increase in body mass or size, for example, we use 300% growth on fingerling trout for a diet study. We need to see 300% growth from the start weight prior to the termination of the study. This provides time for small influences to make real differences.

This said it is clear that Artemia is a poor diet and that amphipods/mixed diets support growth. I'm not sure how your fatty acid data is impacted by sizes for seahorses.

Validity of the findings

I still remain troubled by the fatty acid data quality.

I still question this, I do not find the author's rebuttal to be substantial.

For example:

Captured amphipods

C16:0 29.84, 23.67, 11.21, 25.24
C16:1 0.74, 0.53,13.67, 7.90

16:0 and 16:1 are very diagnostic ratios of fatty acids. within a single treatment "Captured Amphipods" the authors have vastly different ratios. It is hard to believe that animals that are captured from a similar source would have such different ratios. It is more likely that there are some method problems in their fatty acid workup.

The challenge here is that you draw conclusions as to the effects of the FAMES on the PCoA. But your data is highly variable in ways that suggest methods problems. Do you still have your raw data from the GC? Perhaps some separation issues? Again, I really like the idea of the study, but these are serious issues.


If this publication is to proceed, these issues need to be specifically highlighted to the reader.

Additional comments

I still like the idea of the study. There are some real issues with the fatty acid data which are not possible to be addressed in a revision. I'm happy to see if the authors can address these but again argue that the data is seriously challenged.

Reviewer 3 ·

Basic reporting

No special comments, previous issues related to abstract and introduction have been solved by the authors. Just some additional references to be added will be further mentioned in the additional comments section.

Experimental design

Previous issues solved. Perhaps, instead "captured amphipods", a most appropriated term would be "wild" or "wild-captured". Please replace one of the two options along the manuscript.

Validity of the findings

No special comments, previous issued resolved.

Additional comments

Lines referred to attached pdf:
L211. Please clarified: If wet weight was calculated individually but SGR was calculated for each experimental group following the formula presented.
L290. "The Total lipid content was high amongst amphipods from all three sources, with no significant
differences (F = 4.56; p = 0.07) among groups." This sentence should be clarified and rewritten in a more clear way. The lipid content was high compared to "what"? Artemia? please clarify.
L330. The indistinguishable growth rates observed in Seahorses fed in excess with Artemia after almost 2-months experiment results quite curious. Apparently no courtship, gender or pregnancy effect has biased this result, nor prey acceptability as well, makes me think about the protein content of these preys, probably very low, as a potential cue behind this result. Something should be added in the discussion. Other than lipids aminogram or energetic content of each diet could clarify these results.
L447. To add a couple one or two references.

Annotated reviews are not available for download in order to protect the identity of reviewers who chose to remain anonymous.

---

## Round 0.3 · accepted · Accept

This paper has been greatly improved by the review process, and in the last round of reviews, the authors have addressed the concerns of Reviewer 2 related to the statistical methods. I would encourage the authors to make the review documents available to potential readers.